# Rewiring of liver diurnal transcriptome rhythms by triiodothyronine (T₃) supplementation

**Leonardo Vinicius Monteiro de Assis[1]\*[†], Lisbeth Harder[1,2][†][‡], José Thalles Lacerda[3], Rex Parsons[4], Meike Kaehler[5], Ingolf Cascorbi[5], Inga Nagel[5], Oliver Rawashdeh[6], Jens Mittag[7], Henrik Oster[1]\***

[1]Institute of Neurobiology, Center of Brain Behavior & Metabolism, University of Lübeck, Lübeck, Germany; [2]Department of Medical Biochemistry and Biophysics, Karolinska Institute, Stockholm, Sweden; [3]Department of Physiology, Institute of Bioscience, University of São Paulo, São Paulo, Brazil; [4]Australian Centre for Health Services Innovation and Centre for Healthcare Transformation, School of Public Health and Social Work, Faculty of Health, Queensland University of Technology, Kelvin Grove, Australia; [5]Institute of Experimental and Clinical Pharmacology, University Hospital Schleswig-Holstein, Kiel, Germany; [6]School of Biomedical Sciences, Faculty of Medicine, University of Queensland, Brisbane, Australia; [7]Center of Brain Behavior & Metabolism, Institute for Endocrinology and Diabetes – Molecular Endocrinology, University of Lübeck, Lübeck, Germany

**\*For correspondence:**
leonardo.deassis@uni-luebeck.de (LVMdA);
henrik.oster@uni-luebeck.de (HO)

[†]These authors contributed equally to this work

**Present address:** [‡]Division of Molecular Neurobiology, Department of Medical Biochemistry and Biophysics, Karolinska Institutet, Stockholm, Sweden

**Competing interest:** The authors declare that no competing interests exist.

**Abstract** Diurnal (i.e., 24 hr) physiological rhythms depend on transcriptional programs controlled by a set of circadian clock genes/proteins. Systemic factors like humoral and neuronal signals, oscillations in body temperature, and food intake align physiological circadian rhythms with external time. Thyroid hormones (THs) are major regulators of circadian clock target processes such as energy metabolism, but little is known about how fluctuations in TH levels affect the circadian coordination of tissue physiology. In this study, a high triiodothyronine (T₃) state was induced in mice by supplementing T₃ in the drinking water, which affected body temperature, and oxygen consumption in a time-of-day-dependent manner. A 24-hr transcriptome profiling of liver tissue identified 37 robustly and time independently T₃-associated transcripts as potential TH state markers in the liver. Such genes participated in xenobiotic transport, lipid and xenobiotic metabolism. We also identified 10–15% of the liver transcriptome as rhythmic in control and T₃ groups, but only 4% of the liver transcriptome (1033 genes) were rhythmic across both conditions – amongst these, several core clock genes. In-depth rhythm analyses showed that most changes in transcript rhythms were related to mesor (50%), followed by amplitude (10%), and phase (10%). Gene set enrichment analysis revealed TH state-dependent reorganization of metabolic processes such as lipid and glucose metabolism. At high T₃ levels, we observed weakening or loss of rhythmicity for transcripts associated with glucose and fatty acid metabolism, suggesting increased hepatic energy turnover. In summary, we provide evidence that tonic changes in T₃ levels restructure the diurnal liver metabolic transcriptome independent of local molecular circadian clocks.

## Editor's evaluation

de Assis et al. demonstrate a role for T3 in modulating circadian metabolic rhythms both systemically and within the liver. The findings extend the molecular framework in which organismal metabolism is coordinated in a circadian fashion.

**eLife digest** Many environmental conditions, including light and temperature, vary with a daily rhythm that affects how animals interact with their surroundings. Indeed, most species have developed so-called circadian clocks: internal molecular timers that cycle approximately every 24 hours and regulate many bodily functions, including digestion, energy metabolism and sleep.

The energy metabolism of the liver – the chemical reactions that occur in the organ to produce energy from nutrients – is controlled both by the circadian clock system, and by the hormones produced by a gland in the neck called the thyroid. However, the interaction between these two regulators is poorly understood. To address this question, de Assis, Harder et al. elevated the levels of thyroid hormones in mice by adding these hormones to their drinking water.

Studying these mice showed that, although thyroid hormone levels were good indicators of how much energy mice burn in a day, they do not reflect daily fluctuations in metabolic rate faithfully. Additionally, de Assis, Harder et al. showed that elevating $T_3$, the active form of thyroid hormone, led to a rewiring of the daily rhythms at which genes were turned on and off in the liver, affecting the daily timing of processes including fat and cholesterol metabolism. This occurred without changing the circadian clock of the liver directly.

De Assis, Harder et al.'s results indicate that time-of-day critically affects the action of thyroid hormones in the liver. This suggests that patients with hypothyroidism, who produce low levels of thyroid hormones, may benefit from considering time-of-day as a factor in disease diagnosis, therapy and, potentially, prevention. Further data on the rhythmic regulation of thyroid action in humans, including in patients with hypothyroidism, are needed to further develop this approach.

## Introduction

Circadian clocks play an essential role in regulating systemic homeostasis by controlling, in a time-dependent manner, numerous biological processes that require alignment with rhythms in the environment (*Gerhart-Hines and Lazar, 2015*; *West and Bechtold, 2015*; *de Assis and Oster, 2021*). At the molecular level, the clock machinery is comprised of several genes that are organized in interlocked transcriptional-translational feedback loops (TTFLs). The negative TTFL regulators, *Period* (*Per1-3*) and *Cryptochrome* (*Cry1-2*), are transcribed after activation by circadian locomotor output cycles kaput (CLOCK) and brain and muscle ARNT-like 1 (BMAL1 or ARNTL) in the subjective day. Towards the subjective night, PER and CRY proteins heterodimerize and, in the nucleus, inhibit BMAL1/CLOCK activity. This core TTFL is further stabilized by two accessory loops comprised of nuclear receptor subfamily 1 group D member 1–2 (NR1D1-2, also known as REV-ERBα-β) and nuclear receptor subfamily 1 group F member 1–3 (NR1F1-3, also known as RORα-γ), and DBP (albumin D-site binding protein) (*Takahashi, 2017*; *Pilorz et al., 2020*; *de Assis and Oster, 2021*). Upon degradation of PER/CRY, towards the end of the night, BMAL1/CLOCK are disinhibited, and a new cycle starts.

How the molecular clocks in different tissues and downstream physiological rhythms are coordinated has been the subject of increasing scientific interest in recent years. Environmental light is detected by a nonvisual retinal photoreceptive system that innervates the central circadian pacemaker, the suprachiasmatic nucleus (SCN) (*Golombek and Rosenstein, 2010*; *Hughes et al., 2016*; *Ksendzovsky et al., 2017*; *Foster et al., 2020*). The SCN distributes temporal information to other brain regions and across all organs and tissues (*Husse et al., 2015*; *de Assis and Oster, 2021*) through partially redundant pathways, including nervous stimuli, hormones, feeding-fasting, and body temperature cycles. Despite an ongoing discussion about the organization of systemic circadian coordination, all models share the need for robustly rhythmic systemic time cues (*de Assis and Oster, 2021*).

The thyroid hormones (THs), triiodothyronine ($T_3$) and thyroxine ($T_4$), are major regulators of energy metabolism. In the liver, THs regulate cholesterol and carbohydrate metabolism, lipogenesis, and fatty acid (FA) ß-oxidation (*Sinha et al., 2014*; *Ritter et al., 2020*). While circadian regulation of the upstream thyroid regulator thyroid-stimulating hormone (TSH) has been described, $T_3$ and $T_4$ rhythms in the circulation show relatively modest amplitudes in mammals, probably due to their long half-life (*Weeke and Laurberg, 1980*; *Russell et al., 2008*; *Philippe and Dibner, 2015*). Interestingly, in hyperthyroid patients, nonrhythmic TSH secretion patterns are observed (*Ikegami et al., 2019*).

In this study, we investigated how a high $T_3$ state in mice affects diurnal transcriptome organization in the liver. Our data show that tonic endocrine state changes rewire the liver transcriptome in a time-dependent manner independent of the liver molecular clock. The main targets of TH signaling are the genes associated with lipid, glucose, and cholesterol metabolism.

## Results

### Effects of high $T_3$ on behavioral and metabolic diurnal rhythms

We used an experimental mouse model of hyperthyroidism by supplementing the drinking water with $T_3$ (0.5 mg/L in 0.01% BSA). Control animals (CON) were kept under the same conditions with 0.01% BSA supplementation (*Sjögren et al., 2007*; *Vujovic et al., 2015*). TH state was validated by analyzing diurnal profiles of $T_3$ and $T_4$ levels in serum. Significant diurnal (i.e., 24 hr) rhythmicity was detected for $T_3$ in CON with peak concentrations around the dark-to-light phase transition. $T_3$-supplemented mice showed ca. fivefold increased $T_3$ levels compared to CON mice with no significant diurnal rhythm. However, $T_3$ levels showed a temporal variation (ANOVA, p=0.006), which was classified as ultradian by JTK_CYCLE (12 hr period length, *Supplementary file 1*, p=0.01) in the $T_3$-treated group. $T_4$ levels were nonrhythmic in all groups (*Figure 1A, B*, *Supplementary file 1*). Compared to CON, overall $T_4$ levels were reduced two- to threefold in $T_3$-supplemented animals (*Figure 1B*). Resembling the human hyperthyroid condition, $T_3$ mice showed increased average body temperature (*Figure 1—figure supplement 1A*), as well as food and water intake compared to CON mice (*Figure 1—figure supplement 1B and C*). Conversely, $T_3$ mice showed higher body weight on the third week of experimentation (*Figure 1—figure supplement 1D*), as previously shown (*Johann et al., 2019*).

Metabolism-associated parameters such as locomotor activity, body temperature, $O_2$ consumption ($VO_2$), and respiratory quotient (RQ) showed significant diurnal rhythms in both conditions (*Figure 1C–F*, *Supplementary file 1*). No marked differences in locomotor activity were seen between the groups (*Figure 1C*, *Figure 1—figure supplement 1E*). In contrast, in the $T_3$ group, body temperature was elevated in the light (rest) phase (*Figure 1D*, *Figure 1—figure supplement 1F*), leading to a marked reduction in diurnal amplitude. Oxygen consumption in $T_3$ was elevated throughout the day, but this effect was more pronounced during the dark phase (*Figure 1E*, *Figure 1—figure supplement 1G*), leading to an increase in diurnal amplitude. Linear regression of energy expenditure (EE) against body weight in CON and $T_3$ mice (*Tschöp et al., 2012*) revealed no difference in slope, but a higher elevation/intercept was found in $T_3$ mice (*Figure 1—figure supplement 1H*). These data suggest that the higher EE of $T_3$ mice is not only a consequence of increased body weight, but also arises from a higher metabolic state. In $T_3$ mice, RQ was slightly higher in the second half of the dark and the beginning of the light phase, indicating higher carbohydrate utilization during this period (*Figure 1F*, *Figure 1—figure supplement 1I*). In summary, TH-dependent changes in overall metabolic activity were observed resembling the human hyperthyroid condition, albeit with marked diurnal phase-specific effects.

These findings prompted us to evaluate to which extent $T_3$ and $T_4$ levels would be predictive of the overall metabolic state (*TH state effects*) or, alternatively, for changes in metabolic activity across the day (*temporal TH effects*) by correlating hormone levels with metabolic parameters. Correlating the average levels of $T_3$ against activity, body temperature, and $VO_2$ revealed that body temperature and $VO_2$ were positively correlated with $T_3$ levels (*Figure 1G–I*). TH levels and metabolic parameters, however, did not correlate across daytime. Therefore, neither $T_3$ nor $T_4$ qualified as markers for diurnal variations in energy metabolism (*Figure 1J–O*, *Supplementary file 2*). In summary, our data suggest that $T_3$ levels are valid predictors of baseline metabolic state but fail to mirror diurnal changes in metabolic activity at, both, physiological and high $T_3$ states. $T_4$ is an overall poor metabolic biomarker.

### Daytime-independent effects of TH on the liver transcriptome

To study the molecular pattern underlying the observed diurnal modulation of metabolic activity in $T_3$-treated mice, we focused on the liver as a major metabolic tissue. We initially identified time-of-day-independent transcriptional markers reflecting TH state in this tissue. Comparing the liver transcriptome, without taking into consideration the sampling time, 2343 differentially expressed probe sets (2336 genes – DEGs) were identified (±1.5-fold change, false discovery rate [FDR] <0.1, *Figure 2A*, *Supplementary file 3*). Of these DEGs, 1391 and 945 genes were up- or downregulated,

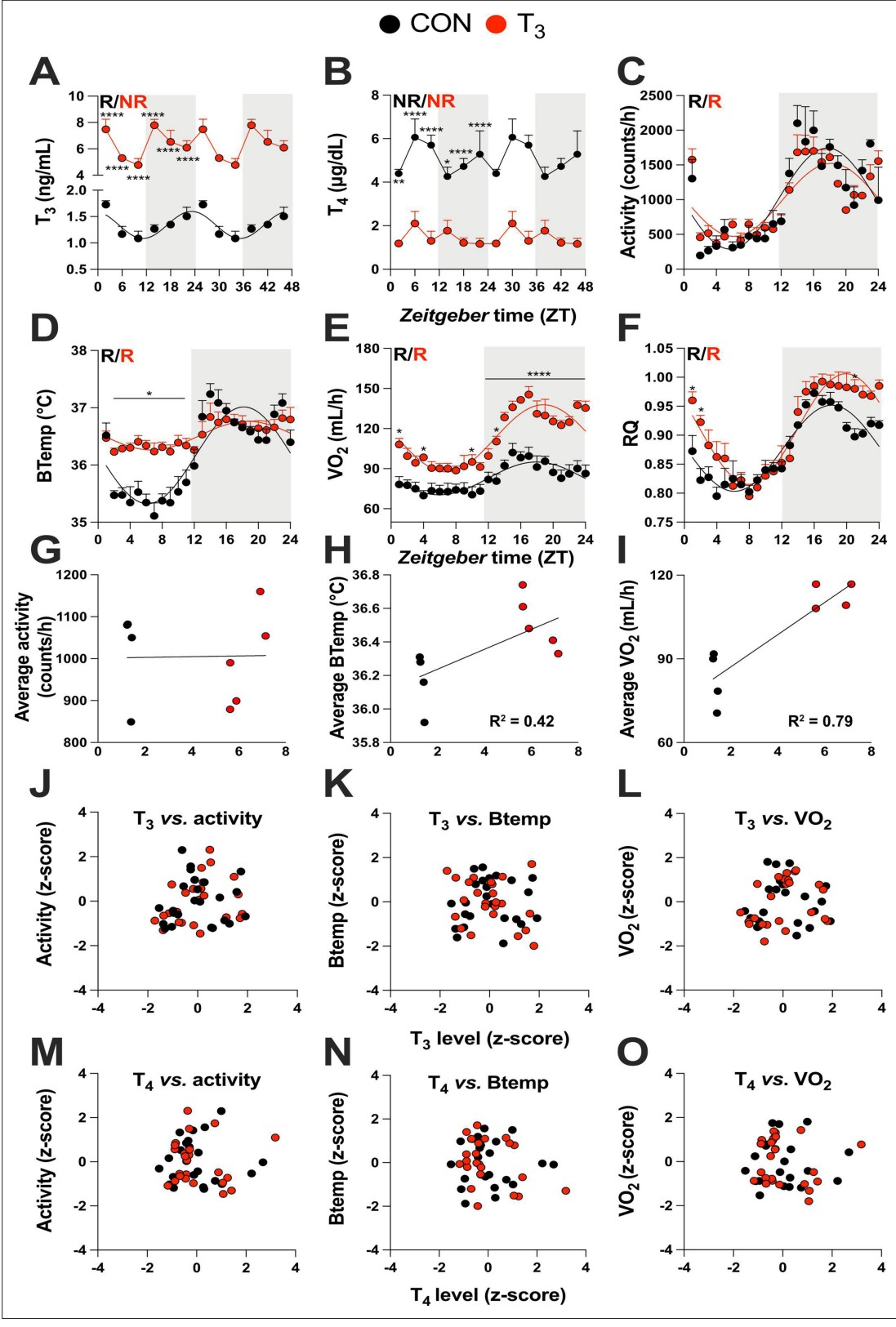

**Figure 1.** Triiodothyronine ($T_3$)-treated mice show classic effects of high thyroid hormone levels compared to control mice (CON). (**A–F**) Serum levels of $T_3$ and thyroxine ($T_4$), 24 hr profiles of locomotor activity, body temperature, $O_2$ consumption, and respiratory quotient are shown. Rhythm evaluation was performed by JTK_CYCLE (p<0.01, **_Supplementary file 1_**). Presence (R) or absence of circadian rhythm (NR) is depicted. In the presence of significant 24 hr rhythmicity, a sine curve was fit. In (**A**) and (**B**), data are double plotted to emphasize the absence or presence of rhythms. (**G–I**) Linear regression

_Figure 1 continued on next page_

*Figure 1 continued*

of T₃ average levels with average of locomotor activity, temperature, and $O_2$ consumption. (**J–O**) Correlation between thyroid hormone levels and normalized levels of metabolic outputs is shown as z-scores (additional information is described in *Supplementary file 2*). In (**A**) and (**B**), n = 4–6 animals per group and/or timepoint. In (**C**) and (**D**), n = 4 and 5 for CON and T₃ groups, respectively. In (**E**) and (**F**), n = 4 for each group.

The online version of this article includes the following figure supplement(s) for figure 1:

**Figure supplement 1.** Metabolic evaluation of control (CON) and triiodothyronine (T₃) mice.

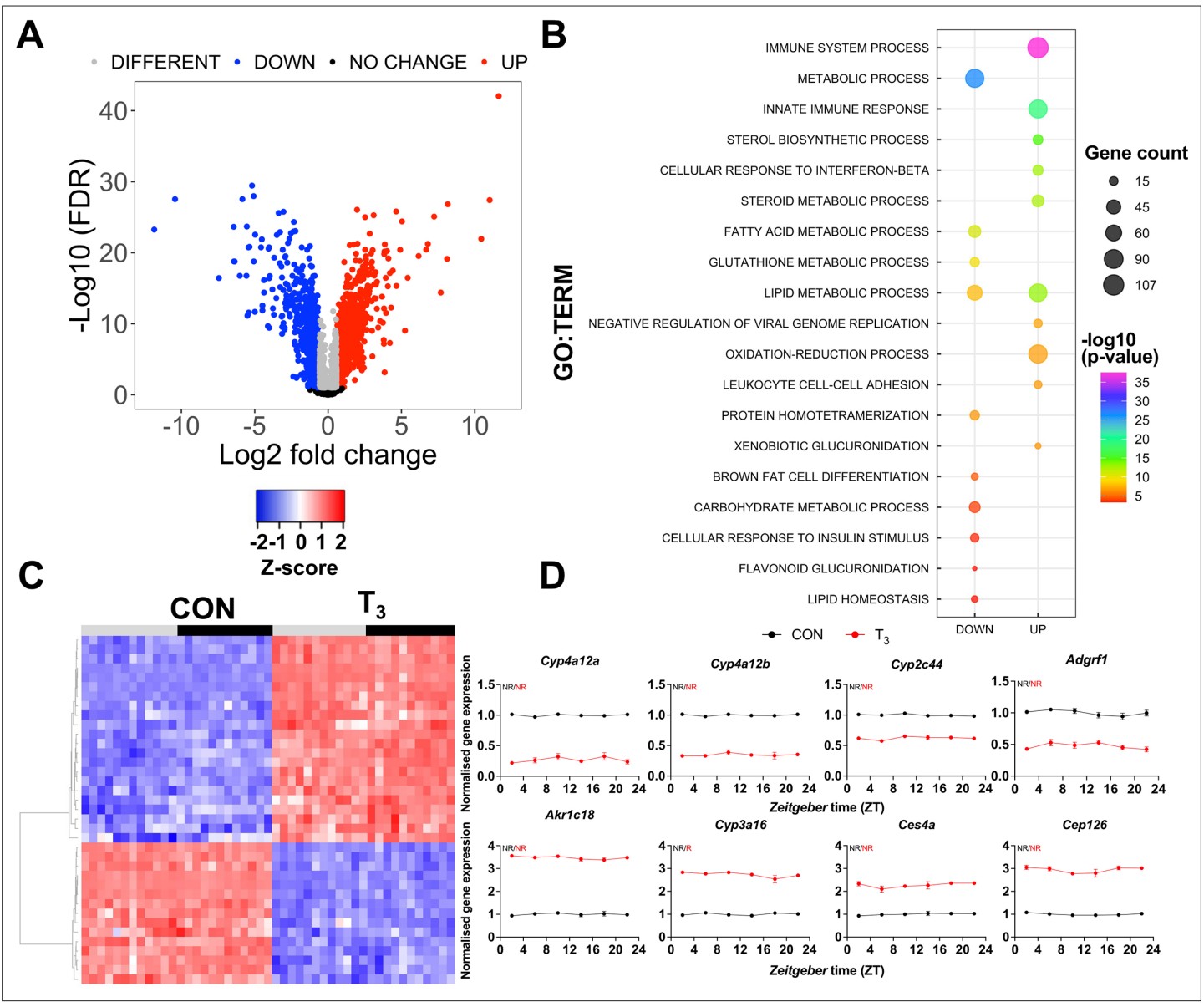

**Figure 2.** Identification of daytime-independent differentially expressed genes (DEGs) in the liver of triiodothyronine (T₃) mice. (**A**) Global (all Zeitgeber times [ZTs] included) evaluation of liver transcriptomes revealed 2336 DEGs of which 1391 and 945 were considered as up- or downregulated, respectively, using a false discovery rate (FDR) < 0.1. Genes with an FDR <0.1 were classified as different irrespectively of fold change values. (**B**) Top 10 list of biological processes from gene set enrichment analysis (GSEA) of up- and downregulated DEGs is represented. Additional processes can be found in *Supplementary file 3*. (**C**) Heatmap of liver DEGs showing significant T₃-dependent regulation across all timepoints. Light and dark phases are shown as gray and black, respectively. (**D**) Diurnal expression profiles of most robustly regulated DEGs. Gene expression of all groups was normalized by CON mesor. Additional information is described in *Supplementary file 4*. None of these genes showed rhythmic regulation across the day (NR). n = 4 samples per group and timepoint, except for the T₃ group at ZT 22 (n = 3).

respectively, by elevated $T_3$ (*Figure 2A*, *Supplementary file 3*). Gene set enrichment analysis (GSEA) of upregulated DEGs yielded processes related to xenobiotic metabolism/oxidation-reduction, immune system, and cholesterol metabolism, amongst others. On the other hand, GSEA of downregulated DEGs yielded biological processes pertaining to FA and carbohydrate metabolism, as well as cellular responses to insulin (*Figure 2B*, *Supplementary file 3*). We identified 37 genes whose expression was robustly up- or downregulated by $T_3$ across all timepoints (*Figure 2C*, *Supplementary file 4*). Genes involved in xenobiotic transport/metabolism (*Abcc3*, *Abcg2*, *Ces4a*, *Ugt2b37*, *Papss2*, *Gstt1*, *Sult1d1*, *Cyp2d12*, *Ephx2*, and *Slc35e3*), lipid, FA, and steroids metabolism (*Cyp39a1*, *Ephx2*, *Akr1c18*, *Acnat1*, *Cyp4a12a/b*, *Cyp2c44*), vitamin C transport (*Slc23a1*), and vitamin $B_2$ (*Rfk*) and glutathione metabolism (*Glo1*) were identified. Additional genes involved in mitosis and replication were also identified (*Cep126*, *Mdm2*, *Trim24,* and *Mcm10*) (*Figure 2D*, *Supplementary file 4*).

We suggest that these transcripts could serve as robust daytime-independent biomarkers of TH state in the liver.

## TH-dependent regulation of liver diurnal transcriptional rhythms

We used the JTK_CYCLE algorithm (*Hughes et al., 2010*) to describe the effects of TH state changes on 24 hr liver gene expression rhythms. We identified 3354 and 2592 probes – comprising 3329 and 2585 unique genes – as significantly rhythmic ($p<0.05$) in CON or $T_3$, respectively (*Figure 3A*, *Supplementary file 5*). Of these, 2319 and 1557 probes were classified as exclusively rhythmic in CON or $T_3$, respectively. A total of 1035 probes (1032 genes) were identified as rhythmic in both groups (*Figure 3A*, *Supplementary file 5*), amongst these most core circadian clock genes (*Supplementary file 5*). Principal component analysis (PCA) showed a distinct pattern of organization across time between the groups for the shared genes (*Figure 3—figure supplement 1*). We next assessed the distribution of phase and amplitude across 24 hr between the groups. Rose plot analyses revealed a similar distribution pattern of phase, but $T_3$ mice showed a higher number of genes peaking in the light phase (Zeitgeber time [ZT] 7–9) and the first half of the dark phase (ZT 13–20) compared to CON (*Figure 3B*). Cross-condition comparison of genes with robust rhythmicity revealed only a minor phase advance of around 1 hr in $T_3$ (*Figure 3C*), which was independently confirmed by qPCR (*Figure 3—figure supplement 2*).

GSEA of rhythmic genes was performed to detect rhythmically regulated pathways under both TH conditions. In CON mice, transport, RNA splicing, lipid and glucose metabolism, and oxidation-reduction processes were overrepresented. In the high $T_3$ condition, several immune-related processes, FA oxidation, and regulation of mitogen-activated protein kinase 1 (MAPK) signaling were found. Interestingly, robustly rhythmic genes were enriched for lipid and cholesterol metabolism and circadian-related processes, suggesting that these processes are tightly coupled to circadian core clock regulation (*Figure 3D*, *Supplementary file 5*). Individual inspection of clock genes revealed the absence of marked effects on mesor (i.e., the midline statistic of the diurnal rhythm sine fit) and amplitude but a slight phase advance (*Figure 3E–F*), which corroborates the phase advance effects seen at the rhythmic transcriptome level (*Figure 3C*).

We next focused on the diurnal regulation of TH signaling by analyzing the expression of genes encoding for modulators of TH signaling, that is, TH transporters, deiodinases, and TH receptors, and established TH target genes. We found that the TH transporter genes, *Slc16a2* (*Mct8*), *Slc7a8* (*Lat2*), and *Slc10a1* (*Ntcp*), lost rhythmicity in $T_3$ mice compared to CON. Amongst the receptors, *Thra* was rhythmic, while *Thrb* was arrhythmic under both conditions. Of the deiodinases, only *Dio1* was robustly expressed under both conditions, but without variation across the day (*Figure 4A*). Significant but nonuniform changes in baseline expression levels were observed for *Slc16a10*, *Slc7a8*, *Dio1* (up in $T_3$) and *Slco1a1*, *Thra*, and *Thrb* (down in $T_3$, *Figure 4A*). To analyze the effect of such changes on TH action, we studied diurnal regulation of established liver TH output genes. Reflecting elevated $T_3$, all selected TH target genes showed increased expression across the day in $T_3$ mice (*Figure 4B, C*). No clear regulation was seen regarding amplitude or phase (*Figure 4C*).

In summary, we provide evidence that the molecular clock of the liver functions independent of TH state. At the same time, changes in diurnal expression patterns were found for FA oxidation- and immune system-related genes in $T_3$ mice. These changes were associated with marked gene expression profile alterations for TH signal regulators and outputs. Collectivity, these data indicate

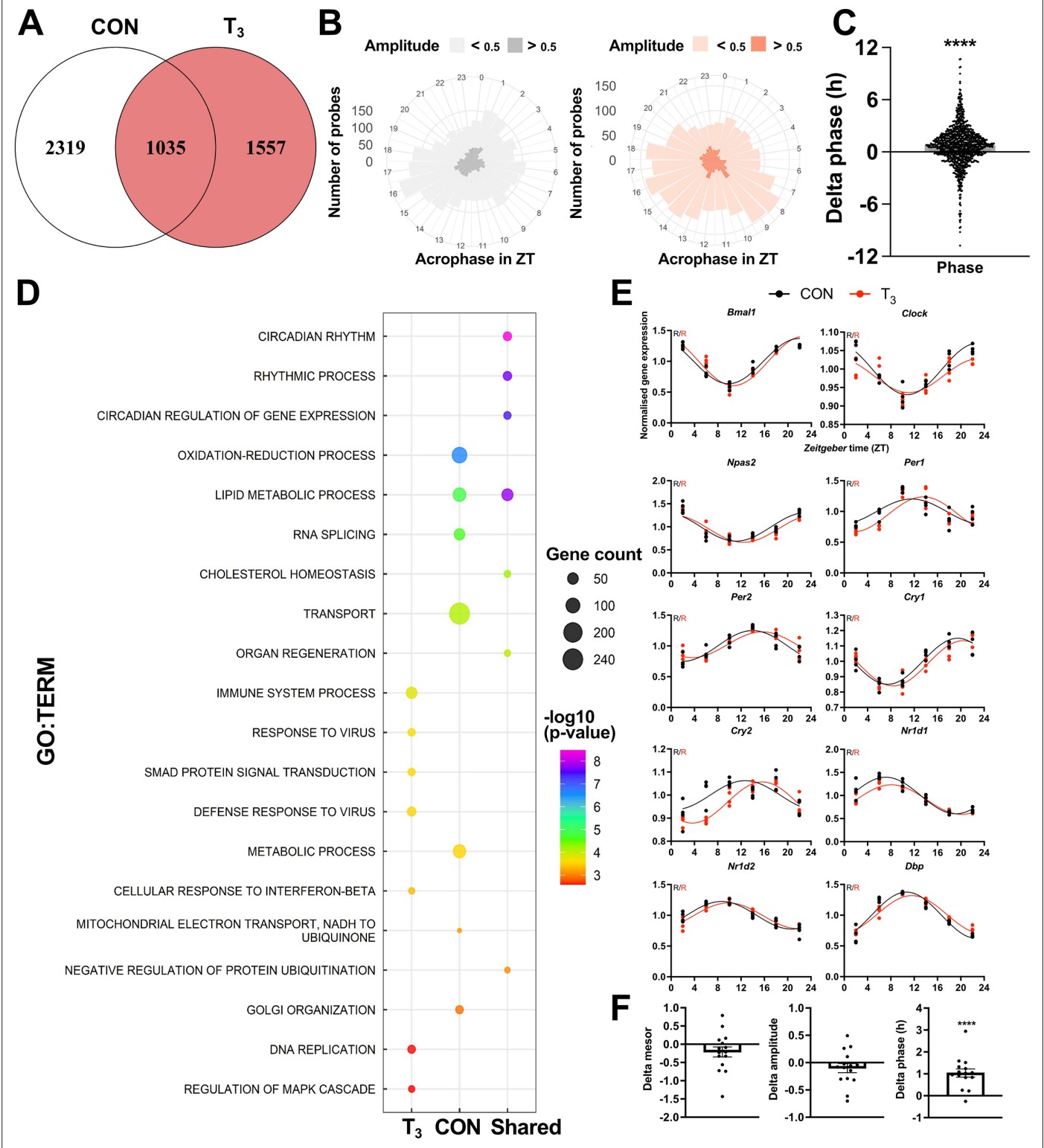

**Figure 3.** Diurnal evaluation of liver transcriptome of triiodothyronine (T₃) mice. (**A**) Rhythmic probes were identified using the JTK_CYCLE algorithm (***Supplementary file 5***). Venn diagram represents the distribution of rhythmic probes for each group. (**B**) Rose plot of all rhythmic genes from control (CON) (gray) and T₃ (red) are represented by the acrophase and amplitude. Phase estimation was obtained from CircaSingle algorithm. (**C**) Phase difference between shared rhythmic genes. Each dot represents a single gene. One-sample *t*-test against zero value was performed and a significant interaction (mean 0.7781, p<0.001) was found. (**D**) Top 7 gene set enrichment analysis (GSEA) of exclusive genes from CON, T₃, and shared are depicted. Additional processes are shown in ***Supplementary file 5***. (**E**) Sine curve was fitted for the selected clock genes. Gene expression of all groups was

*Figure 3 continued on next page*

*Figure 3 continued*

normalized by CON mesor. (**F**) For mesor, amplitude, and phase delta assessment, CircaCompare algorithm was used. The CON group was used as baseline. Additional genes (*Per3*, *Rorc*, *Tef*, *Hif1a*, and *Nfil3*) were used for these analyses. One-sample *t*-test against zero value was used and only phase was different from zero (mean 1.036, p<0.001). n = 4 samples per group and timepoint, except for the $T_3$ group at Zeitgeber time (ZT) 22 (n = 3).

The online version of this article includes the following figure supplement(s) for figure 3:

**Figure supplement 1.** Principal component analysis (PCA) plots of shared rhythmic genes.

**Figure supplement 2.** Validation of clock gene diurnal profile by qPCR.

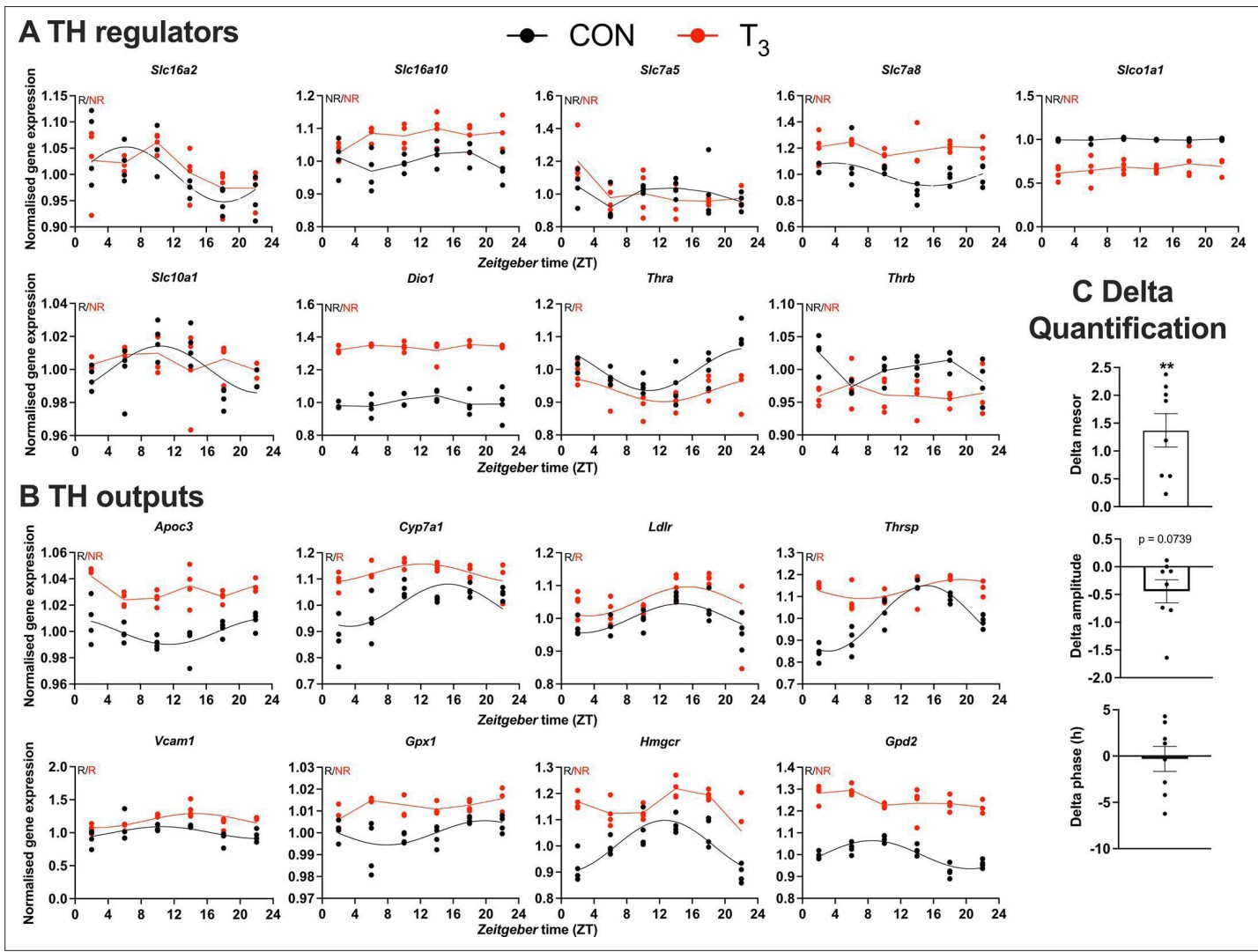

**Figure 4.** Gene expression evaluation of thyroid hormone (TH) regulators and metabolic outputs in triiodothyronine ($T_3$) compared to control (CON). (**A, B**) Genes involved in TH regulation, including transporters, *Dio1*, TH receptors, and well-known $T_3$ outputs are presented. Presence (R) or absence of circadian rhythm (NR) detected by CircaCompare is depicted. Sine curve was fitted for rhythmic genes. Gene expression of all groups was normalized by CON mesor. (**C**) Evaluation of rhythmic parameters from genes described in (**B**) was performed by CircaCompare using CON group as baseline. One-sample *t*-test against zero value was used and only mesor was different from zero (mean 1.371, p<0.01). n = 4 samples per group and timepoint, except for the $T_3$ group at Zeitgeber time (ZT) 22 (n = 3).

an adaptation of the diurnal liver transcriptome in response to changes in TH state in a largely tissue clock-independent manner.

## Quantitative characterization of TH-dependent changes in liver diurnal transcriptome rhythms

To dissect TH state-dependent rhythm alterations in the liver transcriptome, we employed Circa-Compare (*Parsons et al., 2020*) to assess mesor and amplitude in genes that were rhythmic in at least one condition. For precise phase estimation, analyses were performed only on robustly rhythmic genes. Of note, some differences in rhythm classification between JTK_CYCLE and CircaCompare were detected, which is expected due to the different statistical methods. Since we used CircaCompare's rhythm parameter estimations for quantitative comparisons, gene rhythmicity cutoffs in the following analyses were taken from this algorithm. Pairwise comparisons of rhythm parameters (i.e., mesor, amplitude, and phase) revealed predominant effects of TH state on mesor (2519 probes/2504 genes) followed by alterations in amplitude (518 probes/516 genes) and phase (491 probes/genes, *Figure 5A*, *Supplementary file 6*).

We further differentiated CircaCompare outcomes into mesor or amplitude elevated (UP) or reduced (DOWN) and phase delayed or advanced for subsequent GSEA. In these analyses, lipid metabolism was enriched in all categories, except for the phase advance group, which suggests a differential regulation of different gene sets related to lipid metabolism. GSEA of genes with reduced amplitude showed enrichment for FA metabolism and cholesterol biosynthesis, whereas GSEA of elevated amplitude genes showed a strong enrichment for immune system-related genes. Interestingly, genes associated with circadian processes and response to glucose were enriched in the phase delay group (*Figure 5B*, *Supplementary file 6*).

We extracted genes associated with glucose and FA metabolic pathways from KEGG and assessed rhythmic parameter alterations according to CircaCompare (*Figure 5C and E*, *Figure 5—figure supplement 1*). Averaged and mesor-normalized gene expression data of each gene identified by GSEA were used to identify time-of-day-dependent changes in biological processes.

Our data suggest a rhythmic pattern of glucose transport in CON mice roughly in phase with locomotor activity (*Figure 1C*, *Figure 5D*, *Figure 5—figure supplement 1*, *Supplementary file 6*). *Slc2a1* (*Glut1*) was rhythmic in both groups but showed a higher mesor in $T_3$ mice (*Figure 5—figure supplement 1*, *Supplementary file 6*). Conversely, *Slc2a2* (*Glut2*), the main glucose transporter in the liver, was rhythmic in both groups, but it showed a reduced mesor in $T_3$ mice. Other carbohydrate-related transporters such as *Slc37a3* and *Slc35c1* gained rhythmicity and showed higher amplitude and/or mesor in $T_3$ mice. Although GLUT1's role in the liver is minor, increased GLUT1 signaling has been associated with liver cancer and non-alcoholic steatohepatitis (*Chadt and Al-Hasani, 2020*). CON mice, overall rhythmicity in carbohydrate metabolization transcripts, with acrophase in the dark phase, was identified, whereas in $T_3$ mice this process was arrhythmic due to a reduction in amplitude and mesor (*Figure 5D*, *Figure 5—figure supplement 1A*). Individual gene inspection showed that glucose kinase (*Gck*), an important gene that encodes a protein that phosphorylates glucose, thus allowing its internal storing and *Pgk1*, which encodes an enzyme responsible for the conversion of 1,3-diphosphoglycerate to 3-phosphoglycerate, showed reduction in amplitude in $T_3$ mice. Loss of rhythmicity was found for *Pdk4*, a gene that encodes an important kinase that inhibits pyruvate dehydrogenase, and for *Pdhb,* an important component of pyruvate dehydrogenase complex. Reduced inhibition of the pyruvate dehydrogenase complex is known to lead to less glucose utilization via tricarboxylic acid cycle and thus it favors ß-oxidation (*Zhang et al., 2014b*).

Absence of rhythmicity and a higher mesor for the FA biosynthesis rate-limiting gene, *Fasn*, was found in $T_3$ mice, despite this process not being enriched (*Figure 5—figure supplement 1*, *Supplementary file 6*). We identified two subsets of genes with a different regulation at mesor level in FA metabolism (*Figure 5E*). Overall pathway analysis suggested reduced amplitudes associated with a higher mesor. Individual inspection revealed genes mainly related to unsaturated FA, especially with biosynthesis (*Fads2*), and long-chain FA elongation (*Elovl3*, *Acnat1-2*, and *Elovl6*), and oxidation (*Acox2*). *Fads2*, *Elovl2*, and *Elovl3* genes also showed a phase delay. Other subsets of genes showed reduced mesor without changes in amplitude, amongst these genes involved in FA biosynthesis (*Acsm3*, *Acsm5*, *Slc27a2*, and *Slc27a5*), ß-oxidation (*Acaa2*, *Hsd17b4*, *Crot*, *Acadl*, *Acadm*, *Hadh*, *Decr1*, *Cpt1a*, *Acsl1*, and *Hadhb*), glycerolipids biosynthesis (*Gpat4*), and FA elongation (*Hacd3*)

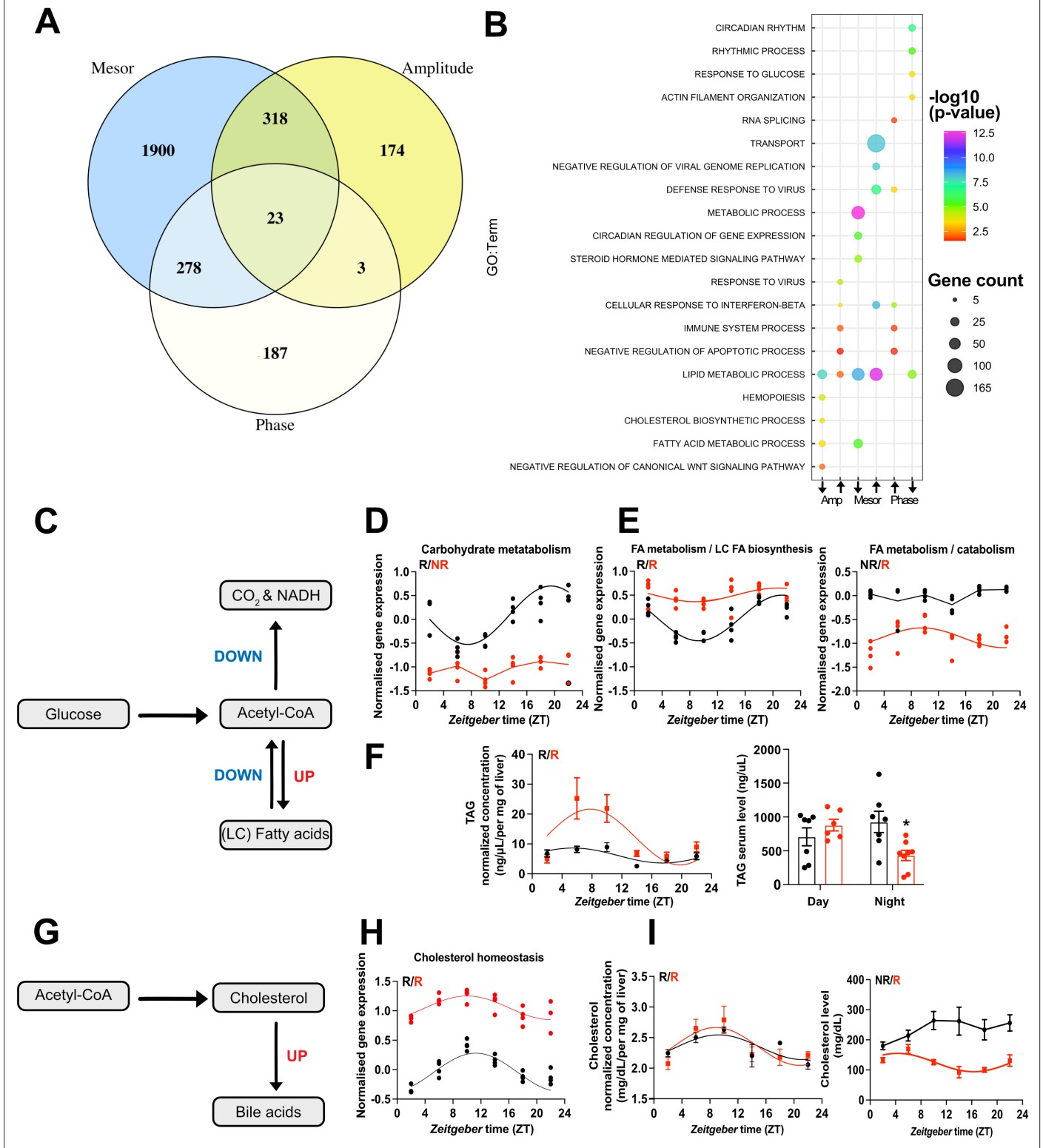

**Figure 5.** CircaCompare analyses of triiodothyronine (T₃) (red) mice compared to control (CON) (black). (**A**) Venn diagram demonstrates the number of probes that displayed differences in each rhythmic parameter (mesor, amplitude, and phase). (**B**) Top 5 enriched biological processes for each rhythmic parameter category. (**C**) Summary of the CircaCompare analyses regarding glucose and fatty acid (FA) metabolism. (**D, E**) Representation of glucose and FA metabolism biological processes obtained from transcriptome data. (**F**) Diurnal rhythm evaluation of liver triacylglyceride (TAG) and day (Zeitgeber time [ZT] 2–6) vs. night (ZT18–22) serum TAG levels comparisons. (**G**) Summary of the CircaCompare analyses regarding cholesterol metabolism.

*Figure 5 continued on next page*

*Figure 5 continued*

(**H**) Representation of cholesterol homeostasis obtained from transcriptome data. (**I**) Diurnal rhythm evaluation of liver and serum cholesterol. Gene expression from each biological process was averaged per ZT and plotted. The reader should refer to the text for detailed information regarding the changes found at the gene level of these processes. Sine curve was fitted for each rhythmic biological process. Individual gene expression pertaining to these processes is found in *Figure 5—figure supplement 1*. n = 4 samples per group and timepoint, except for the $T_3$ group at ZT 22 (n = 3).

The online version of this article includes the following figure supplement(s) for figure 5:

**Figure supplement 1.** Expression profile of the selected genes pertaining to biological processes identified in CircaCompare.

(*Figure 5E*, *Figure 5—figure supplement 1B*, *Supplementary file 6*). To evaluate the metabolic consequences of $T_3$-mediated diurnal rewiring of FA-related transcripts, we measured triacylglyceride (TAG) levels in the liver across the day. TAG levels were rhythmic with an acrophase in the light phase in both groups. However, high $T_3$ levels resulted in a marked increase in amplitude and mesor, thus arguing for a pronounced TAG biosynthesis in the light phase, followed by a stronger reduction in the dark phase, which points to higher TAG consumption. Interestingly, in serum, TAG levels were reduced only in the night phase, likely as a result of the higher energy demands of $T_3$ mice (*Figure 1E*, *Figure 5F*, *Supplementary file 6*). Taken altogether, our data suggest a preferential effect of $T_3$ to increase FA biosynthesis and oxidation and a reduction in glucose metabolization as an energy source in the liver.

A marked diurnal transcription rhythm was observed for cholesterol metabolism genes in CON mice (*Figure 5G and H*). In $T_3$ mice, cholesterol biosynthesis-associated genes were enriched in the amplitude down group, thus suggesting a weakening of rhythmicity. Within this line, the rate-limiting enzyme-encoding gene, *Hmgcr*, showed loss of rhythmicity with reduced amplitude and increased mesor in $T_3$ mice (*Figure 5H*, *Figure 5—figure supplement 1C*, *Supplementary file 6*). Interestingly, upon evaluation of liver cholesterol levels no significant difference was observed, although in both groups, cholesterol levels were rhythmic and with an acrophase in the rest phase. In serum, only in $T_3$ mice cholesterol levels were rhythmic but showed a marked mesor reduction compared to CON, especially in the dark phase (*Figure 5I*, *Supplementary file 6*). Rhythmic genes with a marked higher mesor involved in cholesterol uptake (*Ldlr*, *Lrp5*, and *Nr1h2*) and secretion (*Abcg5*/8 and *Cyp7a1*) in bile acids (*Figure 5G–I*, *Figure 5—figure supplement 1C*, *Supplementary file 6*) were detected in line with $T_3$-mediated increased bile acid production (*Gebhard and Prigge, 1992*; *Bonde et al., 2012*). Taken altogether, our data suggest $T_3$-mediated time-restricted reduction in cholesterol serum levels in favor of increased cholesterol metabolization.

## Discussion

In this study, we analyzed the effects of high $T_3$ state in the mouse liver. Our data argue that $T_3$ is a marker for time-independent metabolic output that is subject to distinct temporal (i.e., diurnal) modulation. At the transcriptome level, $T_3$ induction led to metabolic pathway rewiring associated with only a minor impact on the circadian clock machinery of the liver.

In the human hyperthyroidism condition, $T_3$ and $T_4$ serum levels are both elevated while TSH levels are reduced as part of the inhibitory feedback mechanism of $T_3$ on TSH secretion. In our experimental model, providing $T_3$ levels in the drinking water resulted in the activation of TH effects in the liver, but $T_4$ levels – and, likely, TSH – were reduced, as had previously been reported for this experimental model (*Johann et al., 2019*). Again, this effect is easily explained by the negative feedback of $T_3$ on TSH – and, subsequently, $T_4$ – regulation. Based on this, we refer to our model as '$T_3$ high' instead of hyperthyroid.

Another difference between our experimental model and human hyperthyroidism is an increase in body weight in response to $T_3$ treatment. In fact, some discrepancies between clinical features of hypo- and hyperthyroidism with mouse experimental model for these conditions have been reported (*Johann et al., 2019*; *Kaspari et al., 2020*). In the hypothyroid mouse model, decreased food intake associated with increased EE to maintain core body temperature results in a leaner phenotype (*Kaspari et al., 2020*). For the hyperthyroidism model, although no experimental study has evaluated this discrepancy, it was argued that $T_3$ increases growth hormone biosynthesis promoting higher body weight (*Bargi-Souza et al., 2017*).

Upon analyzing the diurnal metabolic effects of $T_3$, we identified a reduction in core body temperature amplitude due to an elevation in the light phase. Conversely, $T_3$ mice showed a higher $O_2$ consumption amplitude due to increased respiratory activity in the dark phase. Day vs. night analyses confirmed that during the light phase $T_3$ mice have increase metabolic output, which become higher during the dark phase. The absence of an effect in locomotor activity between the groups reinforces the fact of $T_3$ as a strong activator of energy metabolism in our study, which is support by experimental data (*Lanni et al., 2005*; *Cioffi et al., 2010*; *Mullur et al., 2014*; *Jonas et al., 2015*). Thus, one could suggest that several adaptive mechanisms must happen to increase basal metabolic rate. In this line, increased energy output shown by $T_3$ mice seems to rely on a slightly increased glucose (higher RQ quotient) consumption both at light and dark phases. In the liver, our transcriptome analyses revealed important changes in gene expression reflecting increased metabolic output, which will be discussed below.

Although daytime-specific effects in metabolic outputs were observed, no clear correlation between TH levels and metabolic outputs was found, thus ruling out that $T_3$ or $T_4$ are useful *temporal* markers for metabolic output. On the other hand, as a *state* marker, that is, when seen from a longer perspective, $T_3$ served as a robust predictor of metabolic output. For $T_4$, a lack of temporal correlation is easily explained by the absence of diurnal rhythmicity in both normal and high $T_3$ conditions. Conversely, $T_3$ levels were circadian in CON mice.

Previous studies have suggested that serum $T_3$ shows lack of rhythmicity or, if it is present, displays rhythms of small amplitude in humans and/or mice (*Weeke and Laurberg, 1980*; *Russell et al., 2008*; *Philippe and Dibner, 2015*). In our experimental conditions, CON mice displayed a stable diurnal rhythm of $T_3$, albeit with a low amplitude. It should be mentioned, however, that $T_3$ levels showed a temporal variation by ANOVA, with a period of around 12 hr.

Nonetheless, different sets of genes were differentially expressed at different times of the day, thus suggesting time-dependent effects of $T_3$ in the liver. This is suggestive of additional underlying mechanisms that are not dependent on the oscillatory $T_3$ serum levels. We hypothesized that the liver could display increased sensitivity to $T_3$ effects likely via rhythmicity in TH transporters, *Dio1*, and TH receptors expression and/or activity. To illustrate this concept, our transcriptome analyses showed that the liver diurnal transcriptome has 2336 robustly regulated genes (ca. 10% of the transcriptome). Previous studies from the early 2000s using microarrays identified about 2–5% as $T_3$-responsive genes (*Feng et al., 2000*; *Flores-Morales et al., 2002*). Experimental differences such as different $T_3$ levels associated with differences in statistical and significance threshold levels contribute to the differences found between our data and the previous studies. Enrichment analyses showed that elevated levels of $T_3$ were associated with oxidation-reduction and immune system-related genes, whereas a negative association was found for glucose and FA metabolism.

Focusing on comprehending time-of-day-dependent effects in the liver, we focused on the DEGs per timepoint. We identified several hundreds of DEGs across time in $T_3$ mice, thus arguing for a time-dependent effect of $T_3$ in the liver. *Dio1* expression is classically associated with liver thyroid state (*Zavacki et al., 2005*). In our dataset, *Dio1* was differently expressed in all ZTs, except for ZT 22, an effect caused by increased variation in the CON group. This finding may reflect an increased need for $T_3$ metabolization in the liver by DIO1. Although we did not measure DIO1 activity, one could suggest that the observed *Dio1* mRNA upregulation reflects sensitization to a scenario where $T_4$ and $T_3$ are down- and upregulated, respectively. Indeed, DIO1's contribution to thyroid state in the liver is critical while it has little effect on systemic TH levels (*Streckfuss et al., 2005*). Remarkably, 37 genes were identified as time-independent DEGs, that is, displayed stable $T_3$ state-dependent expression across all timepoints, of which were 22 up- and 15 downregulated in $T_3$ mice. These genes participate in several biological processes such as xenobiotic transport/metabolism, lipid, FA metabolism, amongst others. From a translational view, we suggest that these genes could be used to evaluate the thyroid state of the liver at any given time in experimental studies. Moreover, these genes could be used to create a signature of thyroid state in the liver in different conditions and diseases.

While tonic transcriptional targets of $T_3$ have been described in tissues such as the liver, at the same time, robust diurnal regulation of modulators of TH action such as TH transporters, deiodinases, and TH receptors can be observed from high-resolution circadian studies (*Zhang et al., 2014a*; http://circadiomics.igb.uci.edu). This prompted us to study how $T_3$ may affect the transcriptional outputs across the day using established circadian biology methods. Circadian evaluation of CON and $T_3$ livers

revealed 10–15% of the liver transcriptome as rhythmic under both experimental conditions, which is in line with previous experiments (*Zhang et al., 2014a*; *Greco et al., 2021*). A total of 1032 genes (ca. 5% of the liver transcriptome) were robustly rhythmic under both $T_3$ conditions. Overall, the elevation of $T_3$ had a slight phase-delaying effect on these rhythmic genes, which is similar to the effects found in core circadian clock genes. In fact, the similarity in the phase delay between clock gene rhythms and those of robustly rhythmic genes suggests that the latter may indeed involve control through the liver clock. One potential mechanism could involve direct regulation of clock gene transcription by THRB. THRB binding sites are found in the promoter region of several clock genes such as *Bmal1*, *Rev-erbα/β*, *Cry1/2*, and *Per1-3* (GeneCards website, *Safran et al., 2010*). Further experimental studies are required to test this interaction of TH and clock function.

mRNA expression of TH transporter genes, *Slc16a2* (*Mct8*), *Slc7a8* (*Lat2*), and *Slc10a1* (*Ntcp*), showed a loss of rhythmicity while no gain of rhythmicity was found for $T_3$ mice. Such loss of rhythmicity in TH transporters could represent a compensatory mechanism for the higher $T_3$ levels found across the day. The transcriptional response in TH regulators suggests a desensitization mechanism in the liver of $T_3$ mice with a downregulation of TH receptors but increased baseline expression of *Dio1*, *Slc16a10*, and *Slc7a8*. Collectively, these data suggest a compensatory mechanism of decreased signal responses, elevated transport, and metabolization of $T_3$ under high $T_3$ conditions at the mRNA level. However, one must consider the potential diurnal regulation of TH receptor protein levels as well as DIO1 and transporter activity to fully confirm this putative compensatory mechanism.

Regarding diurnal changes, we observed a strong effect of $T_3$ on mesor, followed by changes in amplitude and phase. Interestingly, while circadian parameter analysis revealed a strong effect of $T_3$ on liver transcriptome rhythms, this was mostly without affecting the molecular clock machinery itself. Therefore, $T_3$ effects in the liver seem to act downstream of the molecular clock through a still elusive mechanism. Considering the broad range of changes found in our study, we focused our efforts on comprehending $T_3$ effects on metabolic pathways. Our data reveal a strong $T_3$-mediated diurnal regulation of energy metabolism, mainly related to glucose and FAs, on the mRNA level. Transcripts associated with both processes lost their rhythmicity under high $T_3$ conditions, thus becoming constant across the day. Interestingly, we found evidence that $T_3$ leads to a shift towards FA ß-oxidation over glucose utilization in the liver. $T_3$ effects in FA biosynthesis showed a preferential effect on the synthesis and oxidation of long-chain FA on the mRNA level. Confirming our predictions, livers from $T_3$ mice had higher levels of TAG during the light phase compared to CON, thus suggesting a higher TAG synthesis during the rest phase. However, during the dark phase a marked reduction in TAG serum and liver levels was observed, which suggests an important role of FA ß-oxidation as an energy source to meet the higher energetic demands imposed by $T_3$. Indeed, such changes can be associated with higher energetic demands (higher $VO_2$) both during the light and dark phases in $T_3$ mice. It is a known fact that $T_3$ increases TAG synthesis in the liver (*Sinha et al., 2018*), but our data provide an interesting time of day dependency on $T_3$ effects. Interestingly, no marked alteration in protein catabolism was found, thus suggesting preferential effects of $T_3$ for glucose and FA-related energy sources, at least in the liver (*Mullur et al., 2014*).

Our bioinformatic analyses predicted a higher pool of acetyl-CoA in the liver of $T_3$ mice as a consequence of higher FA ß-oxidation, which we hypothesized to be associated with a putative increased cholesterol biosynthesis. However, no differences were observed in liver cholesterol between the groups, but a marked reduction in serum cholesterol levels was identified in $T_3$ mice. In face of no differences in cholesterol levels in the liver, but associated with a marked reduction in serum cholesterol, we suggest a cholesterol higher uptake and conversion into bile acid. Indeed, such mechanism is supported by our transcriptomic data as well as the literature as $T_3$ is known to increase cholesterol secretion via bile acids or non-esterified cholesterol in the feces (*Mullur et al., 2014*; *Sinha et al., 2018*). Such marked diurnal alterations in the liver transcriptome, especially with regard to metabolic pathways, led us to speculate on the overall consequences of high $T_3$ on organismal rhythms. Loss or weakening of rhythmicity in relevant metabolic processes in other organs, such as the pancreas, white and brown adipose tissue, and other organs, may also take place in the high $T_3$ condition, which could explain the higher energetic demands induced by elevated $T_3$ levels. It is still elusive how $T_3$ affects other metabolic and nonmetabolic organs in a circadian way. Such knowledge will prove useful in designing therapeutic strategies for TH-related diseases such as hepatic steatosis (*Marjot et al., 2022*).

Considering the effects seen in the liver circadian transcriptome, associated with the metabolic data provided, we suggest that $T_3$ may act as a rewiring factor of metabolic rhythms. In this sense, $T_3$ leads to reduction in rhythmicity of major metabolic pathways to sustain higher energy demands across the day. Such pronounced effects are not reflected in marked alterations in the liver clock. From a chronobiological perspective, $T_3$ may be considered a disruptor that uncouples the circadian clock from its outputs, thus promoting a state of chronodisruption (*Potter et al., 2016*; *de Assis and Oster, 2021*). This duality of $T_3$ effects warrants further investigation.

An exciting concept that arises from our data is the concept of chronomodulated regimes for thyroid-related diseases such as hypo- and hyperthyroidism. We suggest evidence that the liver and presumably other organs may show temporal windows in which treatment can be more effective. Based on our diurnal transcriptome data, no optimal time could be suggested due to the lack of rhythmicity for *Dio1*, *Thrb*, and other TH regulator genes. Nonetheless, time-dependent effects in other genes and/or biological processes were identified and could be explored for chronotherapeutic drug intervention.

Taken altogether, our study shows that $T_3$ displays time-of-day-dependent effects in metabolism output and liver transcriptome despite the presence of a strong $T_3$ diurnal rhythm. With regard to metabolism, $T_3$ acts as a *state* marker but fails to reflect temporal regulation of metabolic output. Metabolic changes induced by $T_3$ resulted in a higher overall activation and loss of rhythmicity of genes involved in glucose and FA metabolism, concomitant with higher metabolic turnover, and independent of the liver circadian clock. Our findings reveal a novel layer of regulation of TH action in the liver, and, potentially, in other tissues. As we have shown that high $T_3$ disrupts the rhythmicity of important metabolic processes, this implies that the time of assessment of metabolic parameters in hyperthyroid (and, similarly, hypothyroid) patients should be taken into consideration. Another novel aspect of our findings is the role of liver circadian rhythms in modulating local TH action itself. Such circadian gating might explain variances in the outcomes of $T_3$ treatment if temporal aspects are not taken into consideration. Our data strongly argue for more temporal control in TH studies. Finally, considering that THs are major regulators of metabolism in the liver, therapies of metabolic pathologies could benefit from chronomodulated regimes. Modulation of TH action has been proposed for nonalcoholic fatty liver disease treatment (*Kowalik et al., 2018*; *Zhao et al., 2022*), and our data may assist in the design of a time-of-day-dependent drug regime for this approach (*Marjot et al., 2022*).

One limitation of our finding is the lack of data on the diurnal regulation of $T_3$ effects at the protein level and on enzymatic regulation. It is not unlikely that these represent additional ways by which circadian rhythms and TH action can interact. Our conclusions arise from gene expression data and, therefore, may not fully account for the full spectrum of diurnal modulation of TH action in the livers. Collectively, our data suggest a novel layer of diurnal regulation of liver metabolism that can bear fruits for future treatments of thyroid-related diseases.

# Materials and methods

## Key resources table

| Reagent type (species) or resource | Designation | Source or reference | Identifiers | Additional information |
|---|---|---|---|---|
| Gene (*Mus musculus*) | C57BL/6J | Janvier Labs, Germany | C57BL6JRj | |
| Strain, strain background (C57BL6JRj, male) | C57BL/6JRj | Janvier Labs, Germany | | 2–3-month-old male |
| Biological sample (*M. musculus*) | Liver and serum | Collected and immediately frozen in dry ice | | |
| Sequence-based reagent | RNA extraction | TRIzol, Thermo Fisher Scientific | | |
| Sequence-based reagent | RNA isolation | RNA Miniprep kit Zymo Research | | |
| Sequence-based reagent | cDNA synthesis | High-Capacity Complementary DNA Reverse Transcription Kit, Thermo Fisher | | |
| Sequence-based reagent | qPCR | GoTaq, Promega, USA | | |

*Continued on next page*

*Continued*

| Reagent type (species) or resource | Designation | Source or reference | Identifiers | Additional information |
|---|---|---|---|---|
| Sequence-based reagent | Microarray | WT Plus Kit, Thermo Fisher Scientific | | |
| Sequence-based reagent | qPCR primers | Integrated DNA Technologies (IDT) | | Sequences are provided in the supplementary information |
| Chemical compound, drug | $T_3$ hormone | T6397, Sigma-Aldrich | | |
| Chemical compound, drug | BSA | A7906-50G, Sigma-Aldrich | | |
| Commercial assay or kit | $T_3$ detection kit | DNOV053, NovaTec | | |
| Commercial assay or kit | $T_4$ detection kit | EIA-1781, DRG Diagnostics | | |
| Commercial assay or kit | Triglycerides quantification kit | MAK266, Sigma-Aldrich | | |
| Commercial assay or kit | Cholesterol quantification kit | STA 384, Cell Biolabs | | |
| Software, algorithm | RStudio | R 4.0.3 | | |
| Software, algorithm | Prisma 9 | GraphPad | | |

## Mouse model and experimental conditions

Two- to three-month-old male C57BL/6J mice (Janvier Labs, Germany) were housed in groups of three under a 12 hr light, 12 hr dark (LD, ~300 lux) cycle at 22 ± 2°C and a relative humidity of 60 ± 5% with ad libitum access to food and water. To render mice hyperthyroid (i.e., high $T_3$ levels), the animals received 1 week of 0.01% BSA (Sigma-Aldrich, St. Louis, USA, A7906-50G) in their drinking water, followed by 2 weeks with water supplemented with $T_3$ (0.5 mg/L, Sigma-Aldrich T6397, in 0.01% of BSA). Control animals received only 0.01% BSA in the drinking water over the whole treatment period. During the treatment period, mice were monitored for body weight and rectal temperature (BAT-12, Physitemp, Clifton, USA) individually and food and water intake per cage. All in vivo experiments were ethically approved by the Animal Health and Care Committee of the Government of Schleswig-Holstein and were performed according to international guidelines on the ethical use of animals. Sample size was calculated using G-power software (version 3.1) and is shown as biological replicate in all graphs. Experiments were performed 3–4 times. Euthanasia was carried out using cervical dislocation and tissues were collected every 4 hr. Night experiments were carried out under dim red light. Tissues were immediately placed on dry ice and stored at –80°C until further processing. Blood samples were collected from the trunk, and clotting was allowed for 20 min at room temperature. Serum was obtained after centrifugation at 2500 rpm, 30 min, 4°C and samples stored at –20°C.

## Total $T_3$ and $T_4$ evaluation

Serum quantification of $T_3$ and $T_4$ was performed using commercially available kits (NovaTec, Leinfelden-Echterdingen, DNOV053, Germany, for $T_3$ and DRG Diagnostics, Marburg, EIA-1781, Germany, for $T_4$) following the manufacturer's instructions.

## Serum and tissue TAG and cholesterol evaluation

TAG and total cholesterol evaluation of tissue and serum were processed according to the manufacturer's instructions (Sigma-Aldrich, MAK266 for TAG and Cell Biolabs, San Diego, USA, STA 384 for cholesterol).

## Telemetry and metabolic evaluation

Core body temperature and locomotor activity were monitored in a subset of single-housed animals using wireless transponders (E-mitters, Starr Life Sciences, Oakmont, USA). Probes were transplanted into the abdominal cavity of mice 7 days before starting the drinking water treatment. During the treatment period, mice were recorded once per week for at least two consecutive days. Recordings were registered at 1 min intervals using the Vital View software (Starr Life Sciences). Temperature and

activity data were averaged over two consecutive days (treatment days: 19/20) and plotted in 60 min bins.

An open-circuit indirect calorimetry system (TSE PhenoMaster, TSE Systems, USA) was used to determine respiratory quotient (RQ = carbon dioxide produced/oxygen consumed) and EE in a subset of single-housed mice during drinking water treatment. Mice were acclimatized to the system for 1 week prior to starting the measurement. Monitoring of oxygen consumption, water intake, as well as activity took place simultaneously in 20 min bins. $VO_2$ and RQ profiles were averaged over two consecutive days (treatment days: 19/20) and plotted in 60 min bins. EE was estimated by determining the caloric equivalent according to *Heldmaier, 1975*: heat production (mW) = (4.44 + 1.43 * RQ) * $VO_2$ (mL $O_2$/hr). A linear regression between EE and body weight was performed to rule out a possible confounding factor of body weight (*Tschöp et al., 2012*).

## Microarray analysis

Total RNA was extracted using TRIzol (Thermo Fisher, Waltham, USA) and the Direct-zol RNA Miniprep kit (Zymo Research, Irvine, USA) according to the manufacturer's instructions. Genome-wide expression analyses were performed using Clariom S arrays (Thermo Fisher Scientific) using 100 ng RNA of each sample according to the manufacturer's recommendations (WT Plus Kit, Thermo Fisher Scientific). Data was analyzed using Transcriptome Analyses Console (Thermo Fisher Scientific, version 4.0) and expressed in $\log_2$ values.

## DEG analysis

To identify global DEGs, all temporal data from each group were considered and analyzed by Student's *t*-test and corrected for FDRs (FDR < 0.1). Up- or downregulated DEGs were considered when a threshold of 1.5-fold (0.58 in $\log_2$ values) regulation was met. As multiple probes can target a single gene, we curated the data to remove ambiguous genes. To identify DEGs at specific timepoints (ZTs; ZT0 = 'lights on'), the procedure described above for each ZT was performed separately. Time-independent DEGs were identified by finding consistent gene expression pattern across all ZTs.

## Rhythm analysis

To identify probes that showed diurnal (i.e., 24 hr) oscillations, we employed the nonparametric JTK_CYCLE algorithm (*Hughes et al., 2010*) in the Metacycle package (*Wu et al., 2016*) with a set period of 24 hr and an adjusted p-value (ADJ.P) cutoff of 0.05. For visualization, data were plotted in Prism 9.0 (GraphPad, USA) and a sine wave was fit with a period set at 24 hr. Rhythmic gene detection by JTK_CYLCE was evaluated by CircaSingle, a nonlinear cosinor regression included in the CircaCompare algorithm (*Parsons et al., 2020*), largely (ca. 99%) confirming the results from JTK_CYCLE. Phase and amplitude parameter estimates from CircaSingle were used for rose plot visualizations. To directly compare rhythm parameters (mesor and amplitude) in gene expression profiles between $T_3$ and CON, CircaCompare fits were used irrespective of rhythmicity thresholds. Phase comparisons were only performed when a gene was considered rhythmic in both conditions (p<0.05).

## Gene set enrichment analysis

Functional enrichment analysis of DEGs was performed using the Gene Ontology (GO) annotations for Biological Processes on the Database for Annotation, Visualization, and Integrated Discovery software (DAVID 6.8; *Huang et al., 2009*). Processes were considered significant for a biological process containing at least five genes (gene count) and a p-value<0.05. To remove the redundancy of GSEA, we applied the REVIGO algorithm (*Supek et al., 2011*) using default conditions and a reduction of 0.5. For enrichment analyses from gene sets containing less than 100 genes, biological processes containing at least two genes were included. Overall gene expression evaluation of a given biological process was performed by normalizing each timepoint of CON and $T_3$ by CON mesor. A sine curve was plot and used for representation of significantly rhythmic profiles.

## PCA plots

For PCA, each timepoint was averaged to a single replicate and analyses were performed using the factoextra package in R and Hartigan-Wong, Lloyd, and Forgy MacQueen algorithms (version 1.0.7).

## Data handling and statistical analysis of non-bioinformatic-related experiments

Samples were only excluded upon technical failure. For temporal correlation analyses, normalized values were obtained by dividing each value by the daily group average followed by Z-score transformation. Spearman's correlation analyses were performed between different groups of animals that underwent the same treatment. Analyses were done in Prism 9.0 (GraphPad), and a p-value of 0.05 was used to reject the null hypothesis. Data from ZT0–12 were considered as light phase and from ZT 12–24 as dark phase. Data were either averaged or summed as indicated. Temporal data between groups were analyzed by two-way ANOVA followed by Bonferroni post-test. Single timepoint data were evaluated by unpaired Student's *t*-test with Welch correction or Mann–Whitney test for parametric or nonparametric samples, respectively.

## Data handling and statistical analysis of bioinformatic experiments

Statistical analyses were conducted using R 4.0.3 (R Foundation for Statistical Computing, Austria) or in Prism 9.0 (GraphPad). Rhythmicity was calculated using the JTK_CYLCE algorithm in meta2d, a function of the MetaCycle R package v.1.2.0 (*Wu et al., 2016*). Rhythmic features were calculated and compared among multiple groups using the CircaCompare R package v.0.1.1 (*Parsons et al., 2020*). Data visualization was performed using the ggplot2 R package v.3.3.5, eulerr R package v.6.1.1, and Prism 9.0 (GraphPad). Heatmaps were created using the Heatmapper tool (http://www.heatmapper.ca).

## Acknowledgements

This work was supported by grants from the German Research Foundation (DFG) to HO 353-10/1, GRK-1957, and CRC-296 'LocoTact' (TP13 and TP14). JTH is a fellow of the São Paulo Research Foundation (FAPESP-04524-8/2020). We thank Lucas Moreira Ribeiro, the Federal University of Ouro Preto (UFOP, Brazil), for technical assistance in the bioinformatics analyses.

## Additional information

### Funding

| Funder | Grant reference number | Author |
|---|---|---|
| Deutsche Forschungsgemeinschaft | 353-10/1 | Henrik Oster |
| Deutsche Forschungsgemeinschaft | CRC-296 "LocoTact" (TP14). | Jens Mittag |
| Deutsche Forschungsgemeinschaft | GRK-1957 | Henrik Oster |
| Deutsche Forschungsgemeinschaft | CRC-296 "LocoTact" | Henrik Oster |

The funders had no role in study design, data collection and interpretation, or the decision to submit the work for publication.

### Author contributions

Leonardo Vinicius Monteiro de Assis, Data curation, Software, Formal analysis, Investigation, Methodology, Writing – original draft, Writing – review and editing; Lisbeth Harder, Conceptualization, Data curation, Formal analysis, Investigation, Methodology, Writing – review and editing; José Thalles Lacerda, Rex Parsons, Meike Kaehler, Ingolf Cascorbi, Inga Nagel, Oliver Rawashdeh, Methodology, Writing – review and editing; Jens Mittag, Conceptualization, Writing – review and editing; Henrik Oster, Conceptualization, Supervision, Funding acquisition, Investigation, Writing – original draft, Project administration, Writing – review and editing

### Author ORCIDs

Leonardo Vinicius Monteiro de Assis ⓘ https://orcid.org/0000-0001-5209-0835
Henrik Oster ⓘ http://orcid.org/0000-0002-1414-7068

### Decision letter and Author response

Decision letter https://doi.org/10.7554/eLife.79405.sa1
Author response https://doi.org/10.7554/eLife.79405.sa2

---

## Additional files

### Supplementary files

• Supplementary file 1. Metabolic output rhythmic evaluation by JTK_CYCLE.

• Supplementary file 2. Metabolic output correlation.

• Supplementary file 3. Global differentially expressed genes (DEGs).

• Supplementary file 4. Differentially expressed genes (DEGs) per Zeitgeber time (ZT).

• Supplementary file 5. Rhythmic evaluation by JTK_CYCLE.

• Supplementary file 6. Rhythmic analyses by CircaCompare.

• MDAR checklist

### Data availability

All experimental data was deposited in the Figshare depository (https://doi.org/10.6084/m9.figshare.20376444.v1). Microarray data was deposited in the Gene Expression Omnibus (GEO) database under access code GSE199998 (https://www.ncbi.nlm.nih.gov/geo/query/acc.cgi?acc=GSE199998).

The following datasets were generated:

| Author(s) | Year | Dataset title | Dataset URL | Database and Identifier |
|---|---|---|---|---|
| Assis Lde, Harder L, Oster H | 2022 | Rewiring of liver diurnal transcriptome rhythms by triiodothyronine (T3) supplementation | https://figshare.com/articles/dataset/Rewiring_of_liver_diurnal_transcriptome_rhythms_by_triiodothyronine_T3_supplementation/20376444 | figshare, 10.6084/m9.figshare.20376444.v1 |
| Assis Lde, Harder L, Oster H | 2022 | Rewiring of liver diurnal transcriptome rhythms by triiodothyronine (T3) supplementation | https://www.ncbi.nlm.nih.gov/geo/query/acc.cgi?acc=GSE199998 | NCBI Gene Expression Omnibus, GSE199998 |

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
