## [Editor Report]

de Assis et al. demonstrate a role for T3 in modulating circadian metabolic rhythms both systemically and within the liver. The findings extend the molecular framework in which organismal metabolism is coordinated in a circadian fashion.

---

## [Decision Letter]

**Decision letter after peer review:**

Thank you for submitting your article "Rewiring of liver diurnal transcriptome rhythms by triiodothyronine (T3) supplementation" for consideration by *eLife*. Your article has been reviewed by 3 peer reviewers, one of whom is a member of our Board of Reviewing Editors, and the evaluation has been overseen by a Reviewing Editor and Mone Zaidi as the Senior Editor. The following individual involved in the review of your submission has agreed to reveal their identity: Charna Dibner (Reviewer #2).

Essential revisions:

1) The authors claim on lines 118-121 that control mice show diurnal regulation of T3 levels in Figure 1 A. Furthermore, the authors claim T3 supplemented mice show no diurnal regulation. Figure 1A shows in fact that control mice show low T3 levels with no diurnal regulation while T3 supplemented mice show robust diurnal levels of T3. The data do not support the authors' claims and this needs to be addressed.

2) In Figure 1G-O the authors try to determine if there is a correlation between T3 levels and metabolic parameters. They claim that T3 levels were strongly associated with body temperature and V02 (Figure 1H-I). However, when actual T3 levels were correlated with metabolic parameters (Figure 1J-O) there are no correlations that exist. Once again, the author's claims do not seem to be supported by the data presented. This needs to be clarified how their interpretation of this data was derived.

3) The author's claim to "Comparing the liver transcriptome across times of day and T3 treatment conditions". However, the volcano plot in Figure 2A does not represent a continuum of time. It only shows genes up or down in response to T3 treatment. The time of day that was analyzed is needed for clarification for this figure.

4) In clinical hyperthyroidism, both T3 and T4 are elevated along with decreased TSH levels. It might be helpful to bring up in the Discussion chapter the feedback inhibition of TSH and T4 production upon T3 treatment in the mouse model. Given that T4 is converted into T3 mainly in the liver, the decrease in the substrate (T4) may impact the converting enzyme system (notably, Dio1) in the liver. Did the authors observe these changes in the Deiodinases transcript levels? This point also might be included in the discussion.

5) In Figure 3, There is a weak phase shift in clock gene expression by T3. However, the effect of such change was not observed in other parameters. What is the significance of such changes? The authors should address this within the results or discussion.

*Reviewer #1 (Recommendations for the authors):*

The authors do an elegant job of assessing their goals through comprehensive metabolic and transcriptional profiling. While the authors provide novel insight into the role of T3 in modulating circadian metabolism, both major and minor points arose during the review of this manuscript described below:

1. The authors provide a rich interpretation of their data within the discussion. However, the physiological and clinical relevance of their studies needs to be enhanced. I still have not garnered the importance of these studies and how the field of thyroid disorders is enhanced. This needs to be addressed within the discussion.

---

## [Author Response]

Essential revisions:1) The authors claim on lines 118-121 that control mice show diurnal regulation of T3 levels in Figure 1 A. Furthermore, the authors claim T3 supplemented mice show no diurnal regulation. Figure 1A shows in fact that control mice show low T3 levels with no diurnal regulation while T3 supplemented mice show robust diurnal levels of T3. The data do not support the authors' claims and this needs to be addressed.

We would like to clarify that this apparent discrepancy was contributed by the y-axis scaling. Indeed, in CON group, T_3_ levels are rhythmic (with a period of 24 h), but with a small amplitude. On the other hand, in the T_3_-treated group, T_3_ levels show variations over the course of the day, but with two peaks (around the early morning and early night). Therefore, they are not classified as circadian (i.e., with a period of ca. 24 h), but are considered ultradian according to JTK cycle (Table S1, period of 12h, p = 0.01). In the revised version, a new scaling was introduced, and graphs were double plotted to facilitate visualization. The temporal variations in T_3_ levels were acknowledged.

Line 121 – 124: “However, T3 levels showed a temporal variation (ANOVA, p = 0.006), which classified as ultradian by JTK cycle (12 h period length, Supplementary File 1, p = 0.01) in the T3 treated group. T4 levels were non-rhythmic in all groups (Figure 1A – B, Supplementary File 1).”.

2) In Figure 1G-O the authors try to determine if there is a correlation between T3 levels and metabolic parameters. They claim that T3 levels were strongly associated with body temperature and V02 (Figure 1H-I). However, when actual T3 levels were correlated with metabolic parameters (Figure 1J-O) there are no correlations that exist. Once again, the author's claims do not seem to be supported by the data presented. This needs to be clarified how their interpretation of this data was derived.

Our goal was to distinguish to which extent T_3_ could serve as long-term marker (*trait effect*) and as an acute (“temporal”) marker (*state effect*) of energy metabolism. For the first approach, we compared activity, temperature, and VO_2_ data between the two different T3 conditions. These data indicate that elevation of T3 levels correlates with increased energy turnover in the long run. For the temporal comparison we correlated T3 levels and metabolic parameters at different times of day. To make these differences comparable across the two treatment groups (i.e., to account for trait effects), we normalized all data (T3 and metabolic parameters) to the daily average. Here, all correlative effects are lost. Thus, T3 levels do not predict metabolic activity at any specific time of the day. In other words, T3 is not a *temporal metabolic marker* (Table S2). We think this distinction is very relevant for the field.

We agree with the reviewer’s criticism and in this revised version, the average graphs were removed. We now plot T3 average levels against the metabolic outputs and then a linear regression is performed. In this approach, we could see that average T3 levels correlate with body temperature and VO2.

Line 152 – 156: “Correlating the average levels of T3 against activity, body temperature, and VO2, revealed that body temperature and VO2 were positively correlated with T3 levels (Figure 1G – I). TH levels and metabolic parameters, however, did not correlate across daytime. Therefore, neither T3 nor T4 qualified as markers for diurnal variations in energy metabolism (Figure 1J – O, Supplementary File 2).

3) The author's claim to "Comparing the liver transcriptome across times of day and T3 treatment conditions". However, the volcano plot in Figure 2A does not represent a continuum of time. It only shows genes up or down in response to T3 treatment. The time of day that was analyzed is needed for clarification for this figure.

In the original version, this sentence was meant to indicate that we combined all timepoints and made a global analysis of the DEGs. We have amended the sentence in the revision.

Line 165 – 167: “Comparing the liver transcriptome, without taking into consideration the sampling time, 2,343 differentially expressed probe sets (2,336 genes – DEGs) were identified (± 1.5-fold change; FDR < 0.1; Figure 2A, Supplementary File 3).”

4) In clinical hyperthyroidism, both T3 and T4 are elevated along with decreased TSH levels. It might be helpful to bring up in the Discussion chapter the feedback inhibition of TSH and T4 production upon T3 treatment in the mouse model. Given that T4 is converted into T3 mainly in the liver, the decrease in the substrate (T4) may impact the converting enzyme system (notably, Dio1) in the liver. Did the authors observe these changes in the Deiodinases transcript levels? This point also might be included in the discussion.

Although T_3_-treated mice show classical effects of hyperthyroidism, the T_4_ levels are severely downregulated. This marked reduction on T_4_ level is known to happen in this experimental model and is fully acknowledged in our manuscript. Because of this, we avoided to call T_3_ mie hyperthyroid, but rather we addressed them as T_3_ high.

*Dio1* expression was markedly increased by T_3_ and is among the top 15 highly expressed genes in the global DEGs analysis (Table S3) and it was differentially expressed in every ZT, except in ZT 22 due to a technical variation (Table S4). We did not measure DIO activity, but it is conceivable that the upregulation of Dio1 expression reflects sensitization of the DIO system in the absence of T4. We acknowledged in the manuscript that, although we may see an upregulation of *Dio1*, it is known that this enzyme does not contribute to systemic levels of thyroid hormones, but rather displays an important local role in the liver.

Please find below the new incorporations made into the manuscript.

Line 321 – 327: “In the human hyperthyroidism condition, T_3_ and T_4_ serum levels are both elevated while TSH levels are reduced as part of the inhibitory feedback mechanism of T_3_ on TSH secretion. In our experimental model, providing T_3_ levels in the drinking water resulted in the activation of TH effects in the liver, but T_4_ levels – and, likely, TSH – were reduced, as had previously been reported for this experimental model (Johann et al., 2019). Again, this effect is easily explained by the negative feedback of T_3_ on TSH – and, subsequently, T_4_ – regulation. Based on this, we refer to our model as “T_3_ high” instead of hyperthyroid.”

Line 382 – 386: “This finding may reflect an increased need for T_3_ metabolization in the liver by DIO1. Although, we did not measure DIO1 activity, one could suggest that the observed *Dio1* mRNA upregulation reflects sensitization to a scenario where T_4_ and T_3_ are down- and upregulated, respectively. Indeed, DIO1’s contribution to thyroid state in the liver is critical while it has little effect on systemic thyroid hormone levels (Streckfuss et al., 2005).”

5) In Figure 3, There is a weak phase shift in clock gene expression by T3. However, the effect of such change was not observed in other parameters. What is the significance of such changes? The authors should address this within the results or discussion.

We suggest that T_3_ has a slight, but consistent phase delay on the clock genes. A similar phase delay is found for the robustly rhythmic genes shared between CON and T_3_ groups. The similarity between the phase delay averages might suggest that this is indeed a clock-regulated process. However, it would go beyond the scope of this paper to functionally test this assumption.

We have a statement in the discussion part.

Line 404 – 410: “In fact, the similarity in the phase delay between clock gene rhythms and those of robustly rhythmic genes suggests that the latter may indeed involve control through the liver clock. One potential mechanism could involve direct regulation of clock gene transcription by THRB. THRB binding sites are found in the promoter region of several clock genes such as *Bmal1*, *Rev-erbα/β, Cry1/2*, and *Per1-3* (GeneCards website, (Safran et al., 2010)). Further experimental studies are required to test this interaction of TH and clock function.”

Reviewer #1 (Recommendations for the authors):The authors do an elegant job of assessing their goals through comprehensive metabolic and transcriptional profiling. While the authors provide novel insight into the role of T3 in modulating circadian metabolism, both major and minor points arose during the review of this manuscript described below:1. The authors provide a rich interpretation of their data within the discussion. However, the physiological and clinical relevance of their studies needs to be enhanced. I still have not garnered the importance of these studies and how the field of thyroid disorders is enhanced. This needs to be addressed within the discussion.

We have improved this section in the manuscript.

Lines 481 – 493: “Our findings reveal a novel layer of regulation of TH action in the liver – and, potentially, in other tissues. As we have shown that high T3 disrupts the rhythmicity of important metabolic processes, this implies that the time of assessment of metabolic parameters in hyperthyroid (and, similarly, hypothyroid) patients should be taken into consideration. Another novel aspect of our findings is the role of liver circadian rhythms in modulating local thyroid hormone action itself. Such circadian gating might explain variances in the outcomes of T3 treatment if temporal aspects are not taken into consideration. Our data strongly argue for more temporal control in TH studies. Finally, considering that THs are major regulators of metabolism in the liver, therapies of metabolic pathologies could benefit from chronomodulated regimes. Modulation of TH action has been proposed for non-alcoholic fatty liver disease (NAFLD) treatment (Kowalik et al., 2018; Zhao et al., 2022), and our data may assist in the design of a time-of-day dependent drug regime for this approach (Marjot et al., 2022).